# Unilateral Orbitopathy Caused by Skull Base Chordoid Meningioma

**DOI:** 10.3390/diagnostics13050815

**Published:** 2023-02-21

**Authors:** Jia-He Yang, Meng-Syuan Li, Ming-Jin Shen, Yu-Hsuan Lin

**Affiliations:** 1Department of Otolaryngology, Head and Neck Surgery, Kaohsiung Veterans General Hospital, Kaohsiung 813, Taiwan; 2Department of Ophthalmology, Kaohsiung Veterans General Hospital, Kaohsiung 813, Taiwan; 3Department of Pathology & Laboratory Medicine, Kaohsiung Veterans General Hospital, Kaohsiung 813, Taiwan; 4Institute of Biomedical Sciences, National Sun Yat-sen University, Kaohsiung 804, Taiwan; 5School of Medicine, National Yang Ming Chiao Tung University, Taipei 112, Taiwan; 6School of Medicine, Chung Shan Medical University, Taichung 402, Taiwan

**Keywords:** orbit, skull base, meningioma, endoscopy

## Abstract

Chordoid meningioma (CM) makes up only 1% of all meningiomas. Most cases of this variant are locally aggressive, have high growth potential, and are likely to recur. Although CMs are known to be invasive, they rarely extend into the retro-orbital space. Herein, we report a case of a central skull base CM in a 78-year-old woman whose only manifestation was unilateral proptosis with impaired vision resulting from the tumor extending into the retro-orbital space through the superior orbital fissure. The diagnosis was confirmed by analysis of specimens collected during endoscopic orbital surgery, which simultaneously relieved the protruding eye and restored the patient’s visual acuity by decompressing the oppressed orbit. This rare presentation of CM reminds physicians there may be lesions outside the orbit that can cause unilateral orbitopathy and that endoscopic orbital surgery can be used to confirm its diagnosis as well as treat it.

**Figure 1 diagnostics-13-00815-f001:**
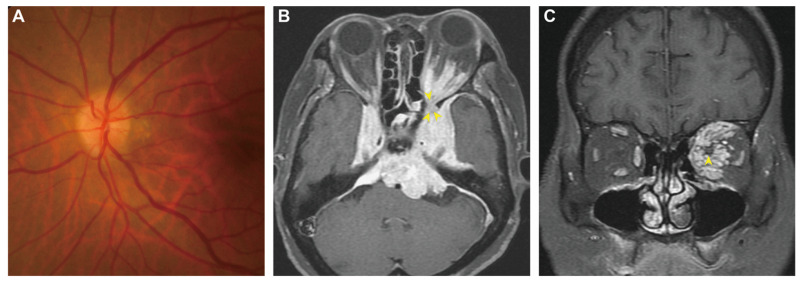
Pre-treatment fundoscopy and brain magnetic resonance imaging (MRI). (**A**) The fundus photo of the left eye showed blurring of the disc margin. (**B**) Axial T1-weighted gadolinium-enhanced MRI showed a vividly enhanced, relatively homogenous skull base lesion extending into the orbit through the superior orbital fissure (yellow arrowheads). (**C**) The coronal view of the T1-weighted sequence showed the tumor grossly infiltrating the intraconal space and encircling the optic nerve (yellow arrowhead). A 78-year-old woman visited our clinic complaining of progressive blurred vision and proptosis of the left eye that had begun two months prior. She reported no previous trauma or additional neurologic deficits. The best corrected left visual acuity was found to be 20/100. Other ophthalmic findings of the left eye included positive relative afferent pupil defect and impaired color vision. Subsequent fundoscopy showed optic disc edema (**A**), confirming left optic neuropathy. Her thyroid blood tests were all within normal limits. Magnetic resonance imaging showed an extensive skull base tumor invading the left retrobulbar space through the superior orbital fissure (**B**,**C**). We then arranged endoscopic endonasal surgery (Figure 2A) to confirm the diagnosis and decompress the orbit, bringing about immediate recovery of eye protrusion, visual acuity, and color perception. Histopathology revealed meningothelial cells with a chordoma-like appearance (Figure 2B–F), confirming the diagnosis of chordoid meningioma (CM). Because curative surgical resection has been associated with high comorbidity, the patient opted to receive Gamma knife radiosurgery (GKRS) instead. The patient remained asymptomatic at 14-month after the definite treatment. Orbital tumors typically originate from the orbit itself, but if they come from the adjacent structures, the most common neoplasms that invade the orbit have been found to be meningiomas from the cranial base [1]. Chordoid meningioma is a rare WHO grade II meningioma that commonly occurs sporadically in females in their forties [2]; however, although uncommon, CM has been linked to Castleman disease [3]. CM biologically behaves with local aggressiveness and destructiveness, high putative growth potential, and a strong propensity for recurrence and malignant transformation [4]. Current evidence demonstrated no significant difference between the typical MR imaging features of CM and other subtypes of meningiomas; both showed hyper-intense signals in the T2-sequence and noticeably homogeneous enhancement after contrast administration. However, one study comparing advanced MRI metrics of eight CMs and 102 WHO grade I-III meningiomas revealed that the apparent diffusion coefficient (ADC) value and normalized ADC ratios of CM were higher than those of other subtypes [5]. A subsequent study additionally found that the proportion of chordoid histology in a whole tumor was positively correlated with its ADC value, suggesting that ADC metrics can be a valuable tool to distinguish CM from different meningioma variants [6]. Once diagnosed, the standard treatment for cranial meningioma was surgical resection [2,4]. Total tumor resection was one of the most crucial prognostic factors, although the intent may cause significant comorbidity. In contrast, the role of radiotherapy (RT) remains debated, with some clinicians prescribing adjuvant radiotherapy in cases where the tumor cannot be completely resected or has high proliferative potential scored by the Mindbomb E3 ubiquitin protein ligase 1 (MIB-1) labeling index [4]. However, a recent review of 221 surgically-treated CM patients demonstrated that adjuvant radiotherapy was not beneficial for disease control [2]. Future well-designed investigations are warranted to clarify the relevance of radiotherapy, particularly in adjuvant RT, for CM patients at high risk of recurrence.

**Figure 2 diagnostics-13-00815-f002:**
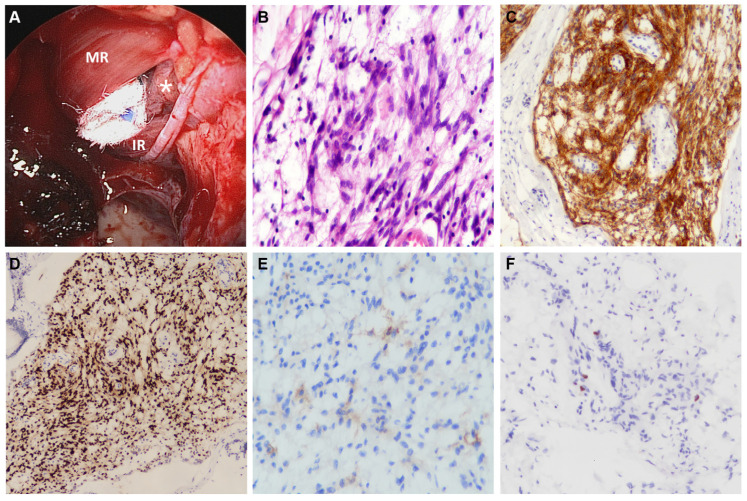
The intraoperative pictures and the histopathological findings. (**A**) The medial rectus muscle (MR) and inferior rectus muscle (IR) were clearly exposed after removing the lamina papyracea and the periorbita. The 0° endoscope revealed the tumoral tissues (asterisk) through the intraconal surgical corridor created between the two muscles. (**B**) The hematoxylin-eosin staining demonstrated syncytial meningothelial cells with eosinophilic cytoplasm arranged in characteristic cords and embedded in a myxoid stroma (original magnification ×400). The immunohistochemistry of the tumor cells was diffusely positive for the epithelial membrane antigen (EMA) (**C**, original magnification ×100) and the progesterone receptor (PR) (**D**, original magnification ×100), and focally positive for somatostatin receptor 2A (SSTR2A) (**E**, original magnification ×400). (**F**) The Mindbomb E3 ubiquitin protein ligase 1 (MIB-1) labeling index of the tumor was 2% (original magnification ×200). We employed transnasal endoscopic orbital surgery (EOS) to retrieve tumoral tissues. The EOS was considered safe and effective because it offered similar or improved surgical visualization and manipulation space than the traditional approach [7]. Additionally, its minimally invasive nature reduces external incisions and associated complications, leading to improved postoperative recovery to facilitate subsequent treatment [7]. Another reason for using EOS in this patient was to relieve the compression on the orbit and optic nerve through the established surgical corridor [8]. In conclusion, this study reminds physicians that unilateral orbitopathy can be caused by lesions beyond the orbit, in this case, CM, which is a rare presentation. It also highlights endoscopic orbital surgery as a minimally invasive approach to diagnose and relieve compressive symptoms in this disease entity. However, close and continuous follow-up is mandatory to evaluate the effectiveness of GKRS in controlling the growth of this aggressive variant of meningioma.

## Data Availability

The data presented in this article are available on request from the corresponding author.

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
