# Peer review of "Unilateral Orbitopathy Caused by Skull Base Chordoid Meningioma"

_diagnostics, 2023, doi:10.3390/diagnostics13050815_

Round 1

Reviewer 1 Report

Dear Prof. Dr. Andreas Kjae, Editor-in-Chief of Diagnostics Journal,

Dear Mr. Dennis Zhu, Managing Editor of Diagnostics Journal,

Dear Authors,

February 6th – 2023, Poland

Re: Peer-Review Report – Minor Revisions

         The article's authors (diagnostics-2230703) discuss a case of "Unilateral Orbitopathy caused by Skull Base Chordoid Meningioma" in a 78-years old female patient who suffered from poor visual acuity and defective color vision due to left eye proptosis as a result of a tumorous extension (invasion) of choroid meningioma to the retro-orbital space. The article belongs to the "interesting images" category. It is well-written for publication and addresses a substantial and rare problem relevant to ophthalmology, oncology, and minimally invasive surgery disciplines. The article conveys high interest for readers, provides a good level of novelty, and emphasizes scientific soundness in describing the methods, study design, and reporting the results. The authors should comply with each element of the peer review comments below. I have also attached the original PDF submission files with highlights and comments to guide the article's revision stage further.

·       The title is good and represents the reported case.

·       The authors should double-check the journal's guidelines for the "interesting images" category at https://www.mdpi.com/journal/Diagnostics/instructions#Images

·       The abstract is reasonable and reflects the diverse domains of the reported case. The authors also adhered to the journal's guidelines concerning the structure of the abstract and the word limit, and they highlighted the rarity, importance, and recommendations for the reported case.

·       The authors should refrain from repeating keywords existing within the title, such as "choroid meningioma". They should also rely principally on terms based on medical subject headings (MeSH) and Emtree keywords.

·       Line #74. The authors should refrain from repeating keywords existing within the title, such as "choroid meningioma". They should also rely principally on terms based on medical subject headings (MeSH) and Emtree keywords.

·       Figure 1 is good, especially concerning the second image (image B), which shows how the tumor infiltrated the left orbital cavity via the supra-orbital fissure.

·       Figure 2 is excellent, but the authors need to explain all abbreviations, including those related to IHC, such as MIB-1 for the Ki-67 receptor expression. Abbreviations can be explained in a footnote below the figure.

·       The references are good, relevant, adequate, and up-to-date. There were no self-citations. Nonetheless, the authors did not rely much on resources of a superior level of evidence, including randomized controlled trials (RCTs), systematic reviews, and meta-analyses. The authors also need to thoroughly check the bibliographic citations while following the journal's instructions for authors concerning the references at https://www.mdpi.com/journal/Diagnostics/instructions#references

Best regards,

The peer-reviewer.

Reviewer 2 Report

This is an interesting case report about a chordoid meninigoma of the central skull base which led to a left orbital proptosis. The left orbita was successfully decompressed with endoscopic endonasal orbital surgery. 

The authors present illustrative figures (intraop and histopathologic) but may

add some relevant histologic features:

EMA positivity was shown but how was staining of Progesterone receptor, SSTR2A, Podoplanin. They are mainly positive in chordoid meningiomas. 

Minor points:

please replace choroid to chordoid meningioma in the text

  •  
